# The Regulatory Role of T Cell Responses in Cardiac Remodeling Following Myocardial Infarction

**DOI:** 10.3390/ijms21145013

**Published:** 2020-07-16

**Authors:** Tabito Kino, Mohsin Khan, Sadia Mohsin

**Affiliations:** 1Cardiovascular Research Center, Lewis Katz School of Medicine, Temple University, Philadelphia, PA 19140, USA; tun60782@temple.edu; 2Center for Metabolic Disease Research, Lewis Katz School of Medicine, Temple University, Philadelphia, PA 19140, USA; mohsin.khan@temple.edu

**Keywords:** regulatory T cells, ubiquitin, mesenchymal stem cell, cortical bone derived stem cell, myocardial infarction

## Abstract

Ischemic injury to the heart causes cardiomyocyte and supportive tissue death that result in adverse remodeling and formation of scar tissue at the site of injury. The dying cardiac tissue secretes a variety of cytokines and chemokines that trigger an inflammatory response and elicit the recruitment and activation of cardiac immune cells to the injury site. Cell-based therapies for cardiac repair have enhanced cardiac function in the injured myocardium, but the mechanisms remain debatable. In this review, we will focus on the interactions between the adoptively transferred stem cells and the post-ischemic environment, including the active components of the immune/inflammatory response that can mediate cardiac outcome after ischemic injury. In particular, we highlight how the adaptive immune cell response can mediate tissue repair following cardiac injury. Several cell-based studies have reported an increase in pro-reparative T cell subsets after stem cell transplantation. Paracrine factors secreted by stem cells polarize T cell subsets partially by exogenous ubiquitination, which can induce differentiation of T cell subset to promote tissue repair after myocardial infarction (MI). However, the mechanism behind the polarization of different subset after stem cell transplantation remains poorly understood. In this review, we will summarize the current status of immune cells within the heart post-MI with an emphasis on T cell mediated reparative response after ischemic injury.

## 1. Introduction

Acute MI is the most severe manifestation of coronary artery disease, which causes more than 2.4 million deaths in the USA, more than 4 million deaths in Europe and Northern Asia [1]. During cardiac ischemic events, the heart undergoes deleterious changes that result in cardiac remodeling of the left ventricular (LV) resulting in both structural and functional alternations. The ischemia in the heart triggers an inflammatory response that leads to the formation of a collagen-rich-scar, which is replaced from necrotic tissue to prevent cardiac rupture. Therefore, it is reasonable to conclude that the healing process is tightly coupled with the inflammatory microenvironment of the infarcted heart [2,3].

The cells of the immune system and their secreted factors play crucial roles in the initiation, progression, and resolution of inflammation following MI. Immune cell subsets contribute to both damage and repair of cardiac tissue specifically in regard to scar formation and LV remodeling [4]. Various types of inflammatory cells are recruited to the damaged area in a temporal fashion, where they remove necrotic tissue and promote scar formation [5]. The participation of T cells in myocardial inflammation and repair has been observed in experimental rodent models. In particular, regulatory T cells (Tregs) mainly mediate organ-specific regenerative programs [6,7,8]. T cell reactivity can benefit myocardial healing by promoting reparative fibrosis in a postmitotic organ [9]. However, sustained T cell responses in the heart can lead to adverse remodeling and contribute to the progression of ischemic heart failure (HF) at later chronic stages [10]. Temporal and spatial regulation from these biphasic immune cell populations is essential to maintain reparative processes [11]. Importantly, focusing on T cells, including Tregs, can be a clue to reveal the reparative mechanism. Moreover, they can be a target of therapy for patients with ischemic heart disease (IHD).

Pharmacotherapy was traditionally promoted in patients with IHD. After surviving from acute coronary syndrome (ACS), optimal medical therapy (OMT) is a golden standard to prevent cardiovascular death [12]. However, OMT cannot promote a regenerative effect in the ischemic area. To date, target therapies are improving and include specific antibodies and the exogenous ubiquitin helping in reducing the scar area in rodent models after cardiac injury [13,14]. In addition, stem cell-based therapies had developed with improvement in cardiac function, however, the overall beneficial effects are relatively modest with fundamental mechanisms of stem cell-mediated repair being largely unknown. This review aims to summarize evidence regarding the role of T cell responses in myocardial remodeling following MI, including how stem cell therapies can be used to mediate the ubiquitination state of T cells.

## 2. Immune Cell Response Post-Ischemic Injury

After MI, the rapid and uncontrolled cellular death and release of intracellular contents into the intercellular space are initiated via necrosis. Necrosis of the ischemic area triggers an inflammatory response in the heart with the infiltration of cells including neutrophils, macrophages, monocytes, T cells, and B cells to clear dead cells and cellular debris [15]. In the first stage, the injured myocardium releases damage associated molecular patterns (DAMPs), which bind toll-like receptors (TLRs), and initiate the production of cytokines/chemokines to induce the activation and recruitment of neutrophils and Ly6C^high^ monocytes. Some Ly6C^high^ monocytes differentiate into M1 macrophages, which contain a pro-inflammatory secretome enriched in interleukin (IL)-1β, tumor necrosis factor (TNF)-α, and IL-6 [11]. In the second stage, Ly6C^low^ monocytes and M2-like macrophages with high expression of IL-10, transforming growth factor (TGF)-β and vascular endothelial growth factor (VEGF) come in with the anti-inflammatory function much needed after injury [16]. After this phase, fibroblasts are activated and move into the infarcted area. Cytokines/chemokines and growth factors play a critical role in the differentiation of fibroblasts into myofibroblasts [17]. Myofibroblasts, a major producer of extracellular matrix (ECM) proteins, produce matrix metalloproteinases (MMPs) and tissue inhibitors of metalloproteinase (TIMPs). Therefore, a highly regulated balance between MMPs and TIMPs is essential in the maintenance of ECM homeostasis [18]. Figure 1 shows the mechanisms of the innate and adaptive immune response after cardiac damage and tissue repair.

Monocytes and macrophages have a reparative effect in ischemic heart tissue after MI. As their activation mainly occurs in acute phase, other components also regulate repair mechanism from acute to chronic phase. The adaptive immune response, especially T cells, participate in those cascades including myofibroblasts transition. The adverse cardiac remodeling may occur when there is an inappropriate regulation of inflammation. However, the mechanism of adverse cardiac remodeling is unclear and is linked to chronic inflammation. There is a dire need of accumulated studies in chronic phase to illustrate cardiac remodeling following MI including immune response to better understand the wound healing response with potential clinical application.

## 3. The Role of the Adaptive Immune Response Focused on T Cells after Cardiac Damage

CD4+ T helper (Th) cells provide proper immune cell homeostasis and host defense but CD4+ T cells have also been shown to promote autoimmune and inflammatory diseases. The original description of Th cells included Th1 and Th2 cells; however, additional Th cell subsets were discovered later, including Th17 and Tregs, which are characterized by specific cytokine profiles [19]. In the healthy myocardium, there are about 10^3^–10^4^ CD4+ T cells per heart assessed by a flow cytometry analysis [20]. Based on histological observation, CD4+ T cells accumulate in the infarct zone within 2 min after reperfusion [21]. In a permanent coronary occlusion injury model, infiltrating CD4+ T cells influx gradually and peak at day 7. Th1 cells and Tregs are the predominant subsets of CD4+ T cells, whereas Th2 cells and Th17 cells are minor populations. CD8+ T cells, γδT cells, and B cells also peaked on day 7 post-MI [22]. In clinical studies, patients with ACS have significantly greater activation of Th1 and Th17, and a prominent decrease in Treg numbers [23,24].

During scar maturation, the inflammatory response enters the chronic phase, which is characterized by low-grade, persistent inflammation. Few studies have reported the chronic inflammatory response after MI. One pilot study describes a global expansion of CD4+ T cells, especially Th2 and Th17 subsets, and a small number of CD8+ T cells at 8 weeks post-MI. This study also showed that depleting CD4+ T cells prevented adverse ventricular remodeling [10]. Another study demonstrated that tissue-specific T-cell responses with predominantly Th1 cells and cytotoxic CD8+ T cells phenotypes are present in ischemic failing human hearts, which contribute to the progression of HF from T cell receptor (TCR) sequencing [25]. According to these studies, CD4+ and CD8+ T cells participate in chronic stage after cardiac injury, but further studies need to be performed to reveal the regulation of adaptive immune responses.

IL-17A producing Th17 effector cells have an important role after MI. Th17 cells can be primed through activation of conventional type 2 dendritic cells (DCs) after MI [26]. IL-17A is primarily produced by γδT cells in the infarcted heart [27] and are involved in late remodeling stages after MI by promoting a sustained infiltration of neutrophils and macrophages and upregulation of pro-inflammatory cytokines leading to cardiomyocyte death and fibrosis [28] via the MMP/TIMP signaling pathway [29]. In a study using IL-17A deficient mice, there was no difference in infarct size compared with WT mice at day 1. However, improved survival and attenuated LV dilation were observed over 28 days post MI [28]. Additionally, several reports showed that IL-17 can directly activate MMP-1 in human cardiac fibrosis via p38 mitogen activated protein kinase (MAPK) and extracellular regulated kinase (ERK) dependent activator protein 1, nuclear factor (NF)-κB activation [30]. Additionally, increased expression of downstream target genes including IL-6, TNF, chemokine ligand (CCL) 20 and C-X-C motif chemokine ligand (CXCL) 1 were also upregulated [31]. In clinical studies, both circulating Th17 cells and IL-17 levels are increased in patients with ACS and stable angina compared to healthy controls [32,33]. Most studies conclude that a high IL-17A circulating level is associated with poor prognosis. One study, however, found that low serum levels of IL-17 are associated with a higher risk of major cardiovascular events in patients with acute MI [34]. Conclusively, the role of Th17/IL-17A in the context of MI has not yet been adequately addressed.

CD8+ T-cells also have been involved in both beneficial and detrimental cardiac remodeling. CD8+ T cells, which have angiotensin II receptors, infiltrate the peri-infarct myocardium 7 days after MI [35]. CD8+ T cells were characterized by upregulated IL-10 and downregulated IL-2 and INF-γ production, which have been shown to reduce ischemic heart injury. On the other hand, another study reported CD8+ T cells activation eliciting determinantal effects on cardiomyocytes in vitro [36]. The cytotoxic activity against healthy cardiomyocytes was myocyte-specific, which suggested major histocompatibility complex (MHC) class I and an antigen-specific cytotoxic response. CD8+ T cells may be detrimental to the cardiac remodeling process by inducing direct cytotoxic effects on healthy cardiomyocytes and amplifying neutrophil and macrophage-mediated inflammation, thus resulting in increased LV dilation and decreased cardiac function. However, CD8+ T cell actions would be indirectly beneficial by decreasing MMP-mediated collagen turnover, facilitating scar formation and decreasing incidence of cardiac rupture [37]. Moreover, CD8+ T cells can reduce cardiac fibrosis and improve cardiac function after injury in mice by the ablation of fibroblast activation protein [38].

Adaptive immune response focused on T cells play an important role to regulate inflammation from the acute to chronic phase. However, the detailed inflammation mechanism remains to be investigated as each T cells subsets are dynamic. Many more studies are needed to correlate the role of different T cell subset dynamics in regard to species, age, genetic modifications, injury models, and postoperative days. Conducting cell-specific transcriptome or proteome analyses of the temporal dynamics of cardiac immune cell accumulations following MI may contribute to identify the mechanism of inflammation after cardiac tissue damage including cytokines/chemokines. Moreover, we must corroborate rodent models in a clinical setting to benefit patients with cardiac injury. Clinically relevant studies are always challenging due to inherent variability in between the sample and tissue availability at multiple time points after injury. Most of the clinically relevant data is on human cells that are usually acquired from blood samples, so they are not necessarily a reflective of ischemic heart tissue conditions. However, currently, we can use human tissue cells under specific environments including ischemic cardiac injury from the cell bank, which can help us solve a complicated puzzle of the wound healing process after MI.

## 4. The Role of the Adaptive Immune System in Cardiac Tissue Repair

After the inflammatory phase, dying neutrophils are cleared by local macrophages that switch their phenotypic polarization to support healing [39]. Tregs can be a potential solution for tissue repair as they can terminate the pro-inflammatory phase and initiate the anti-inflammatory or pro-reparative phase at the site of tissue injury. Natural Tregs are generated in the thymus during fetal development and the first few years of life, while induced Treg cells can be developed later in the periphery from naive CD4+ T cells. Tregs express a specific transcription factor called FoxP3, for the forkhead/winged helix transcription factor, crucial for their development and functions [40]. Few resident Tregs are present in the healthy myocardium, but they can rapidly infiltrate after acute ischemia and have been shown to peak by day 7 after MI [41,42]. Tregs can suppress effector activities of differentiated CD4+ T cells, CD8+ T cells, Th17 cells, and the function of natural killer and B cells [43]. They also influence wound healing processes by modulating monocyte/macrophage differentiation. Some studies have suggested that TGF-β plays a critical role in the induction of FoxP3 expressions and is a main regulator of Tregs, in vivo and in vitro [44].

The ablation of Tregs with CD25 antibody enhances the numbers of both inflammatory myeloid cells and lymphocytes associated with M1 macrophage polarization and delay the healing response [41]. Tregs also promote the differentiation of recruited Ly6C^high^ monocytes toward anti-inflammatory M2 macrophages in the myocardium by secreting IL-10, IL-13, and TGF-β, which can directly stimulate fibroblasts in the myocardium as shown in Figure 1 [45]. Tregs also induced the expression of mediators in macrophages such as osteopontin and transglutaminase factor XIII; these factors are well known to contribute to myocardial healing [36]. Tregs attenuate myocardial ischemia/reperfusion (I/R) injury through a CD39 dependent mechanism involving Akt and ERK pathway [46], and modulated matrix-preserving cardiac fibroblast phenotype, reducing expression of α-smooth muscle actin and MMP-3, and attenuating contraction of fibroblast-populated collagen pads in the later phase [47]. Based on these findings, Tregs play an essential role in cardiac tissue repair.

Tregs have a potential to be a main target to improve cardiac function in ischemic heart tissue, however, reparative ability of Tregs decrease during aging [48]. It is important to investigate factors that can increase the life of a reparative Tregs. Treg capacity in neonatal rodent models is considered much higher than adults, but not fully explored. Therefore, focusing on rejuvenating factors identified from neonatal models may become a clue to develop our understanding of the cardiac repair mechanism in elderly patients.

## 5. The Role of the Adaptive Immune System Focused on T Cells Associated with Ubiquitin

Ubiquitin is important for regulating protein turn over via the ubiquitin–proteasome system (UPS). The UPS regulates fundamental cell functions including mitosis, DNA replication and repair, cell differentiation, transcriptional regulation, and receptor internalization, which all play a role in heart biology [49]. UPS is not only important in protein degradation, but it is also involved in multiple inflammatory processes such as NF-κB activation [50]. In the heart, several ubiquitin ligases (E3 enzymes) are involved in cardiac physiology, such as atrogin 1, muscle RING finger family (MuRF) 1, and murine double minute (MDM) 2 [49]. Consequently, UPS regulates cardiac signal transduction pathways and transcription factors such as calcineurin, which are associated with regulation of pathological hypertrophic growth exhibited post-MI.

Regarding T cells, ubiquitination plays a crucial role in the regulation of TCR-proximal signaling. It also regulates the initial activation and subsequent differentiation of T cells [51], TCR-stimulated endocytosis, and degradation of the linker for activation of T cells (LAT); although the underlying mechanism is poorly defined [52,53].

There are few studies associated with ubiquitin and T cells directly. Recently, however, some studies implied that exogenous ubiquitin participated in cardioprotective function with Tregs via C-X-C motif chemokine receptor (CXCR) 4. Exogenous ubiquitin plays a protective role in attenuating the cardiac inflammatory response and decreasing infarct size after I/R injury. Moreover, exogenous ubiquitin increased the expression of MMP-2 and MMP-9, which can increase ECM degradation and can contribute to a reduction in infarct size [54]. Extracellular ubiquitin interacts with CXCR4 and affects the proliferation of cardiac fibrosis via the ERK 1/2 pathway [55,56]. Another study revealed that a CXCR4 antagonist promotes tissue repair after MI by enhancing Treg mobilization and immune-regulatory function in mice [57]. In a clinical study, combined computed tomography and a positron emission tomography can detect CXCR4 as a surrogate for T cells in humans after MI [9].

There is no direct evidence of exogenous ubiquitin of Tregs via CXCR4. Moreover, exogenous ubiquitin was not illustrated in the T cell ubiquitination even if it is a component of the UPS. However, exogenous ubiquitin has a therapeutic potential and a clue to identify the mechanism between the UPS and the Treg phenotype/recruitments. Future accumulated studies about CXCR4 may contribute to adapt evidence of rodent models to humans.

## 6. Stem Cell Derived Therapies for Cardiac Injury

Current therapeutic and interventional options for the treatment of acute MI focus on the prevention or reverting adverse cardiac remodeling [58,59]. Current therapeutic options can help with functional output, however, it cannot regenerate the lost myocytes due to an ischemic episode. It is now well established that cardiomyocytes in the mature heart cannot proliferate much due to their limited self-regenerative capacity [60]. Exogenous stem cells therapies gain prominence in the last decade due to their ability to repair damaged organs. However, the mechanism by which the organ function improves is debatable and involves minimal regeneration of myocyte warranting investigation of other reparative processes responsible for improvement in function. Other biological processes that can be involved in the wound healing process are immune cells that are one of the key players in a wound healing process after injury. Recently, significant advances have been made in MI treatment using mesenchymal stem cells (MSCs) due to their angiogenic, antiapoptotic, anti-inflammatory, and cardioprotective effects [61,62,63,64]. MSCs have been tested for their ability to differentiate into several lineages in vitro and also tested in animal models to mediate tissue repair [61]. They have been reported to exert profound immunoregulatory effects on DCs, Tregs, and monocytes or macrophages via paracrine effects [65,66,67,68,69].

The potential for MSCs to induce and increase proliferation of Tregs has been shown via a wide range of credible direct and indirect mechanisms. A direct contact between human MSCs and purified CD4+ T cells has been shown to be essential for the induction of Treg as elimination of contact by a semipermeable membrane greatly compromises FoxP3 expression [70]. Soluble factors such as TGF-β1 are a potent paracrine factor that induce FoxP3 expression [71]. A significant increase in the number of FoxP3+ Treg was observed when human CD4+ T cells were cocultured with dental pulp MSCs, which was in turn reduced when TGF-β1 production was blocked [72]. Recently, MSCs have been shown to promote the immune regulatory effects through the release of extracellular vesicles (EVs) and has been suggested as a mechanism of the Treg proliferation. As exosomes payload is rich in mRNA, miRNA, and protein cargo, and have the potential to regulate immune cell gene transcription, intracellular signaling, and effector function [73]. MSC-EV also conditioned human DCs and has shown to have increased secretion of IL-10 and TGF-β leading to greater Tregs induction in pancreatic islet antigen-specific stimulation assays of T cells with type 1 diabetes [74].

Similarly, a novel stem cell population isolated from the cortical bone has demonstrated improved cardiomyocytes survival, cardiac function, and attenuation of remodeling compared with other stem cells because their secretome is enriched in pro-reparative factors [75]. Additionally, CBSCs also show unique expression of immunomodulatory proteins and cytokine production, which affect cardiomyocytes and immune cell populations. Intramyocardial injections of CBSCs resulted in CBSC engraftment and a decrease in the frequency of apoptosis overall 7 days after MI [76]. Concurrently, T cell and macrophage recruitment to the tissue was increased in CBSC treated animals. CBSCs produce cytokines and growth factors that are known to promote T cell and macrophage growth, chemotaxis, differentiation, and survival such as: TIMP-1, CXCL12, and macrophage colony stimulating factor (M-CSF). CXCL12, as a ligand of CXCR4 [77], strongly attracts lymphocytes [78] and has also been shown to protect cardiomyocytes from apoptosis [79]. M-CSF is known to induce the differentiation of bone marrow cells to macrophages, induce an immunosuppressive phenotype, and induce the production of CCL2 [80]. CCL2 is a potent chemotactic signal molecule for monocytes and lymphocytes, including T cells [81], and plays a crucial role in healing infarcts [82].

Clinical applications have demonstrated the ability of bone marrow-derived cells [83,84,85], cardiac-derived cells [86,87], and MSCs [88,89,90] to offer moderate functional benefits when transplanted after cardiac injury. Although, they show sustained modest beneficial effects and bring new knowledge for the cardiac repair mechanism, the basic biology and interactions of stem cells with the host cells and its immune environment are still not fully understood and need to be studied in details. We must identify the key regulators of wound healing processes that are delivered or activated after cell therapy. Moreover, we should focus on the time of delivery of stem cells as the delivery of stem cell factors in a timely manner can be a key for modulating the immune response that effects the wound healing process.

## 7. Conclusions

Accumulative evidence has shown that the adaptive immune response is involved in post-ischemic cardiac remodeling after MI. From the acute to chronic phase, T cells have played an important role in the mediation of tissue repair following cardiac injury. Tregs can terminate the pro-inflammatory phase and initiate the anti-inflammatory or regenerative phase, promoting the differentiation of Ly6C^high^ monocytes toward M2 macrophages in the myocardium by secreting pre reparative cytokines including IL-10, IL-13, and TGF-β. They also stimulate fibroblasts directly in the myocardium. Paracrine signaling to and from immune cells and stem cells can be key in understanding the wound healing process after cardiac injury. Stem cells can induce Tregs via direct and indirect mechanisms. However, the mechanism behind polarization of different T cell subsets after stem cells transplantation remains poorly understood. If we focus on the UPS and its effect on cell therapy, it can give us some clues on how to modulate a pro-reparative phenotype especially associated with Tregs. This may potentially unravel some mechanisms that can augment cardiac healing after ischemic injury. The development of new therapeutic strategies targeting the adaptive immune system in IHD and via stem cells and its interplay with the UPS can contribute to be a more effective treatment in patients with heart disease.

## Figures and Tables

**Figure 1 ijms-21-05013-f001:**
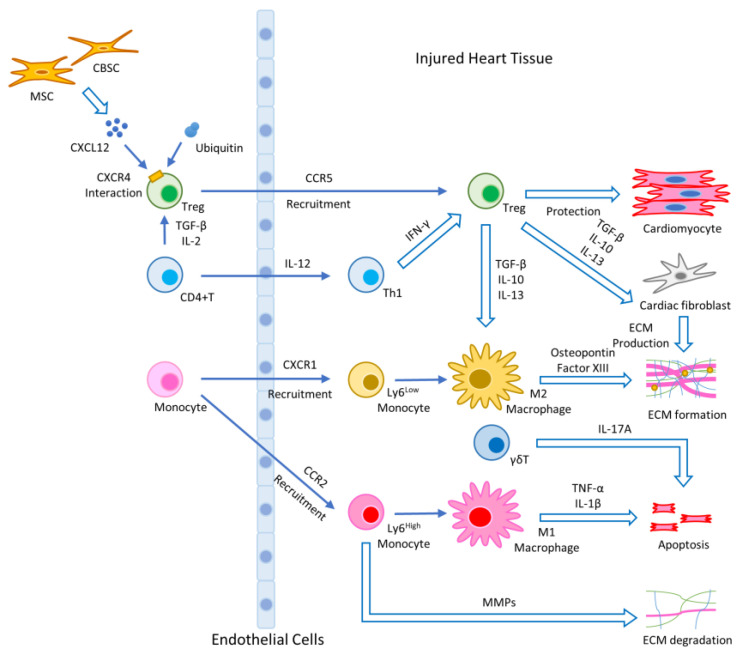
Illustration of the wound healing process after stem cell therapy: activated monocytes/macrophages produce various chemokines and cytokines that initiate inflammation and cell migration. Chemokines produced at the site of infarction induce CCR2-dependent migration of proinflammatory Ly6C^high^ monocytes, which secrete pro-inflammatory cytokines and produce a MMP to degrade ECM. Ly6C^high^ pro-inflammatory monocytes differentiate into classically activated M1 macrophages that express IL-1β and TNF-α. The migration of anti-inflammatory Ly6C^low^ monocytes is mediated via CXCR1. Ly6C^low^ reparative monocytes can differentiate into alternatively activated M2 macrophages. MI induces the activation and proliferation of CD4+ T cells in the heart-draining lymph nodes, which express high levels of IFN-γ at the site of infarction. Additionally, γδT cells participate in apoptosis. Tregs produce IL-10, IL-13, and TGF-β and play an important role in the resolution of inflammation and cardiac repair following MI. MSCs and CBSCs produce chemokine. CXCL12 is a ligand of CXCR4, and exogenous ubiquitin interacts with CXCR4. They participate in cardioprotective function with Tregs.

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
