# Peer review of "The Regulatory Role of T Cell Responses in Cardiac Remodeling Following Myocardial Infarction"

_ijms, 2020, doi:10.3390/ijms21145013_

Round 1
Reviewer 1 Report
This is a scholarly review describing the role of the adaptive immune system in cardiac remodeling following myocardial infarction. The authors sequentially discuss: 1) immune response events upon ischemic injury; 2) the role of the adaptive immune response after cardiac damage and in cardiac tissue repair; 3) the importance of ubiquitin-proteasome pathway in ischemic inflammation mediated by T cells, and 4) stem cell-derived therapies for cardiac injury.
The review is merely descriptive and lacks author's view on the problem. I think it would benefit from adding the drawbacks of the existing experimental models and future perspectives. This can be placed at the end of each section or as a separate section. In other words, review must contain the constructive criticism of the models we have now and suggestions to improve them. Another major point is clinical relevance of these models and the translational pitfalls, which should also be discussed.
Author Response
We thanks the reviewer for constructive feed back.
Comment from Reviewer: I think it would benefit from adding the drawbacks of the existing experimental models and future perspectives. This can be placed at the end of each section or as a separate section. In other words, review must contain the constructive criticism of the models we have now and suggestions to improve them. Another major point is clinical relevance of these models and the translational pitfalls, which should also be discussed.
Response: As recommended by the reviewer, we have added new text highlighted in blue throughout the manuscript to add our comments, concerns and suggestion in the review and it is placed at the end of each section as per reviewer suggestion. We hope that the reviewer will find our response appropriate.

Reviewer 2 Report
Interesting review on T cell mediated responce after MI.
the paper describe a note and intriguing role of immune cell in mechanisms related to cardiac remodellind after myocardial infarction. In particular the paper summarize the possible role of T cell mediated reparative responce
myocardial infarction such as inflammatory dilated cardiomyopathy and valvular diseases such as aortic valve stenosis are characterized by chronic inflammation mediated mainly by T lymphocytes and the associated enhancement of reactive fibrosis. Thus, inflammation can take 2 paths (the inhibition or promotion of fibrosis), depending on the phase of inflammation, inducing pathological cardiovascular remodeling. Elucidation of the regulatory mechanisms of inflammation and fibrosis will contribute to the development of new therapeutic approaches for cardiovascular diseases.
The role of modulation of the immune response in the pathology of myocardial infarction and in post-infarction remodeling involved in one of the most intriguing challenges of modern cardiology
Author Response
We thank the reviewer for their feedback and we really appreciate that they find our review interesting.
Round 2
Reviewer 1 Report
The authors well addressed my comments; the paper can be accepted in a present form.